# Artificial Diets Modulate Infection Rates by *Nosema ceranae* in Bumblebees

**DOI:** 10.3390/microorganisms9010158

**Published:** 2021-01-12

**Authors:** Tamara Gómez-Moracho, Tristan Durand, Cristian Pasquaretta, Philipp Heeb, Mathieu Lihoreau

**Affiliations:** 1Research Center on Animal Cognition (CRCA), Center for Integrative Biology (CBI), CNRS, University Paul Sabatier, 31062 Toulouse, France; tristan.durand@protonmail.com (T.D.); cristian.pasquaretta@univ-tlse.fr (C.P.); mathieu.lihoreau@univ-tlse3.fr (M.L.); 2Laboratoire Evolution et Diversité Biologique, UMR 5174 Centre National de la Recherche Scientifique, Université Paul Sabatier, ENSFEA, 31062 Toulouse, France; philipp.heeb@univ-tlse3.fr

**Keywords:** *Bombus terrestris*, *Nosema ceranae*, nutrition, experimental infection, survival analyses, PCR

## Abstract

Parasites alter the physiology and behaviour of their hosts. In domestic honey bees, the microsporidia *Nosema ceranae* induces energetic stress that impairs the behaviour of foragers, potentially leading to colony collapse. Whether this parasite similarly affects wild pollinators is little understood because of the low success rates of experimental infection protocols. Here, we present a new approach for infecting bumblebees (*Bombus terrestris*) with controlled amounts of *N. ceranae* by briefly exposing individual bumblebees to parasite spores before feeding them with artificial diets. We validated our protocol by testing the effect of two spore dosages and two diets varying in their protein to carbohydrate ratio on the prevalence of the parasite (proportion of PCR-positive bumblebees), the intensity of parasites (spore count in the gut and the faeces), and the survival of bumblebees. Overall, insects fed a low-protein, high-carbohydrate diet showed the highest parasite prevalence (up to 70%) but lived the longest, suggesting that immunity and survival are maximised at different protein to carbohydrate ratios. Spore dosage did not affect parasite infection rate and host survival. The identification of experimental conditions for successfully infecting bumblebees with *N. ceranae* in the lab will facilitate future investigations of the sub-lethal effects of this parasite on the behaviour and cognition of wild pollinators.

## 1. Introduction

Bees face a large diversity of parasites and pathogens that negatively affect their physiology [1,2,3], behaviour, and cognition [4], altogether compromising the fitness of individuals and colonies [5]. The microsporidia *Nosema ceranae* is one of the most prevalent parasites of honey bees worldwide and a major cause of colony declines [2]. *N. ceranae* invades the epithelial cells of the honey bee midgut where it replicates [6,7]. At the physiological level, the parasite disrupts the carbohydrate [8,9] and lipid [10] metabolisms of the host. This causes energetic stress leading to increased consumption of sucrose solution [11,12,13], and alteration of gene expression in the brain [14], inhibition of the apoptosis of epithelial cells [15], and deregulation of immune responses [2]. At the behavioural level, infected honey bees start foraging earlier in life [16,17,18], exhibit more frequent but shorter foraging flights [19,20,21], show reduced homing abilities [22], and display lower olfactory learning performances [23,24].

Recently, *N. ceranae* has also been identified in wild bee species (bumblebees [25,26,27], stingless bees [28,29]) as well as in some wasps [28] and butterflies [30]. Horizontal transmission to wild pollinators may occur through the contamination of flower pollen by infected honey bees. Parasite spillover is particularly concerning since many of these pollinators are solitary or live in small colonies [31], and therefore lack social immunity behaviours that allow honey bees to limit infection risks and combat parasites [32].

The effects of *N. ceranae* on wild pollinators have been best investigated in bumblebees. Recent studies suggest that *N. ceranae* impairs the cognitive abilities of bumblebees, potentially reducing the foraging performances of entire colonies [23,33]. However, the diversity of experimental infection protocols used in earlier studies, and their relatively low success rates (i.e., from 0% in [33,34] to 66% in [35]) make it difficult to draw definitive conclusions and call for more robust standardised approaches. By contrast, 100% of infections are routinely reached in honey bee studies [36].

Published experimental infections with *N. ceranae* in bumblebees followed protocols applied in honey bees, but with no standardisation yet (see summary in Table 1). The age of insects, their level of starvation before exposure, the number of parasite spores to which they are exposed, and the way to inoculate them are all potentially important parameters that vary across current infection protocols [37,38]. Bumblebees were always exposed individually, and generally *N. ceranae* spores were provided in a solution delivered with a micropipette, although other approaches have been tested [39]. Parasite doses ranged from a few thousand to millions of spores per bee. Starvation duration varied from 30 min to 8 h and different ages were tested. Temperature conditions also varied from 25 or 33 °C. These different attempts led to contrasting results, especially concerning the infection rates. Thus far, only two studies achieved rates higher than 50% [27,35] on the basis of molecular analyses (Table 1). Longevity, as a measure of virulence of *N. ceranae*, is also highly variable with bumblebee survival rates ranging from 38% [27] to 95% [34] after 15 days of infection.

Another potentially important parameter, thus far unexplored, is the nutritional composition of diets provided to bumblebees after parasite exposure. Nutrition is a key mediator of host–parasite interactions [40] and should therefore be carefully controlled when developing standard infection procedures. Many insects increase their consumption of dietary protein to develop stronger immunological responses and combat parasites [41,42,43]. Increasing evidence indicates that bees given a choice of foods adjust their intake of nutrients to reach target levels maximising fitness traits. Although there is no direct demonstration that diet modifies the ability of bees to fight infections, bumblebees fed a low-protein diet have a reduced immune response [44]. Honey bees infected by *Nosema apis* [45] and *N. ceranae* [46,47,48] survived longer when provided protein rich pollen. Since protein consumption is needed for synthesising peptides in immune pathways [41,42,49], these results suggest that bees can adjust their nutrient intake for self-medication [50].

Here, we developed an experimental protocol to efficiently infect bumblebees with *N*. *ceranae* by briefly exposing individuals to controlled amounts of parasite spores and feeding them with artificial diets varying in their protein to carbohydrate ratios. We calibrated our approach by testing two spore dosages and two diets. We analysed parasite prevalence (proportion of infected bumblebees based on polymerase chain reactions, PCR), parasite intensity (spore counts in the gut), and host survival to identify the best conditions for infecting bumblebees at sub-lethal doses. To further analyse parasite infection, we analysed the faeces of bumblebees over 21 days in search of *N. ceranae* spores as evidence of parasite multiplication in the host. Finally, we performed a histological study of the ventriculi of bumblebees (where *N. ceranae* is found in honey bees [6,51]) in search of parasite forms inside the epithelial cells.

## 2. Materials and Methods

### 2.1. Bees

We ran 3 experiments with bumblebee workers (*Bombus terrestris*) of unknown age from 4 commercial colonies (Biobest, Belgium: experiment 1: 2 colonies, experiment 2: 1 colony, experiment 3: 1 colony). Before the experiments, we verified the absence of *N. ceranae* and other common parasites of bumblebees (*N. bombi* and *Crithidia bombi*) from a sample of 15 workers of each colony [27]. Diagnoses were performed in monoplex PCRs with the primers 218MITOC [52], Nbombi-SSU-J [53], and CB-ITS1 [54].

We used honey bees as a positive control of infection in experiment 1. *Apis mellifera* workers from unknown age came from a colony in our experimental apiary (University Paul Sabatier—Toulouse III, France) that tested free for *N. ceranae* and *N. apis* in a duplex PCR (primers 218MITOC and 321APIS, [52]), from a sample of 15 workers.

### 2.2. Parasites

We obtained *N. ceranae* spores from naturally infected honey bee colonies (*Apis mellifera*) at our experimental apiary. These colonies were spatially segregated from the *Nosema*-free colonies used for controls (see Section 2.1). We prepared spore solutions from samples of 15 honey bees as follows. We extracted the gut of the bees that we crushed in 15 mL of dH_2_O. Once the presence of spores was verified using light microscopy (×400), we checked the homogenates in a duplex PCR [52] to verify the presence of *N. ceranae* and the absence of *N. apis* (as spores of both parasites have similar morphologies; [55]). We then purified homogenates following a standard protocol [56]. Briefly, 1 mL of the honey bee gut homogenate was centrifuged at 5000 rpm for 5 min. The supernatant containing tissue debris was discarded and the pellet containing the spores was re-suspended in 0.5 mL of dH_2_O by vortexing. The sample was centrifuged and washed into distilled H_2_O two times more to obtain a spore solution of 85% purity [56]. To prepare the inoculum, we counted *N. ceranae* spores using an improved Neubauer haemocytometer [57] in a light microscope (×400). Each sample was counted twice, and the total number of spores was averaged. We adjusted the final inoculum concentration to either 7500 spores/μL or 15,000 spores/μL in 20% (*v*/*v*) of sucrose solution. Spore solutions and inoculums were prepared no more than 1 week before the infections.

### 2.3. Infections

The day of infection, we isolated bumblebees in Petri dishes and starved them for 5 h. Afterwards, inside the Petri dish, we delivered a drop of 20 μL sucrose solution (20%; *w*/*v*) of the corresponding inoculum, containing either 150,000 (i.e., 150 K) or 300,000 (i.e., 300 K) spores, while control bumblebees only received sucrose solution. To make sure that all bumblebees were exposed to the same parasite dose, we only used bumblebees that drank the whole drop of sucrose in a maximum time of 2 h.

We followed the same protocol to infect honey bees (experiment 1), except that honey bees were starved only for 2 h, as preliminary observation showed this time is enough to elicit feeding. We coordinated both bumblebee and honey bee groups to perform the parasite exposure at the same time in both groups. Exposed honey bees received 150 K spores, and control 0 spores.

### 2.4. Artificial Diets

Diets were liquid solutions containing either a low protein to carbohydrate ratio (P/C 1:207.4, hereafter “low-protein diet”) or a high protein to carbohydrate ratio (P/C 1:6.2, hereafter “high-protein diet”). The two diets contained a fixed total amount of nutrients (P + C content of 170 g/L). Carbohydrates were supplied as sucrose (Euromedex, Strasbourg, France). Proteins consisted of a mixture of casein and whey (4:1) (Nutrimuscle, Paris, France). Both diets contained 0.5% of vitamin mixture for insects (Sigma, Darmstadt, Germany). Diets were delivered to bees in a gravity feeder, which consisted of a 1.5 mL Eppendorf tube with a hole at its basis through which the bees could insert their proboscis and ingest liquid food.

### 2.5. Experiment 1. Does Diet Influence N. ceranae Infection?

We tested the influence of diet and parasite dose on *N. ceranae* infection by analysing survival, parasite prevalence, and parasite loads in bumblebees. We used 300 bumblebee workers from 2 colonies. Bumblebees were exposed either to 150,000 (150 K) or 300,000 (300 K) spores of *N. ceranae*. Control individuals received only sucrose solution. Once the bumblebees had drunk the entire drop of sucrose (i.e., 272 out of 300 individuals), we allocated them to 1 of the 2 artificial diets, thereby generating 6 experimental groups of 44 to 47 individuals each (Table 2). We kept these bumblebees individually in a Petri dish with a hole (1 cm Ø) on the top lid, in which we placed the gravity feeder containing 1 of the 2 diets. We replaced feeders with new feeders containing a fresh diet every day. We maintained the bumblebees in 2 identical incubators (Pol-Eko, Wodzisław Śląski, Poland) at 26 °C, with a 12 h light/12 h dark photoperiod for 21 days. Each incubator contained the same proportion of bees from the 6 experimental groups. The 9% of bumblebees that did not drink the sucrose solution were excluded from the experiment.

#### 2.5.1. Survival

To account for survival, every day we recorded the number of dead bees and stored them at −20 °C for later analyses. At the end of the experiment, we freeze-killed all surviving bumblebees and stored them at −20 °C.

#### 2.5.2. Parasite Prevalence

We analysed the presence of *N. ceranae* in the gut of bumblebees by PCR (see example Figure 1A). We extracted the gut of each bumblebee and homogenised it in dH_2_O. We vortexed a fraction of this homogenate with 2 mm glass beads (Labbox Labware, Barcelona, Spain) to break the parasite spores, and extracted genomic DNA using Proteinase K (20 mg/mL; Euromedex, Strasbourg, France) and 1 mM of Tris-EDTA buffer (pH = 8). In every round of extraction, we included a sample containing *N. ceranae* spores as a positive control. We amplified DNA with the primers 218MITOC specific for *N. ceranae* [52]. PCR reactions were carried out in 48-well microtitre plates in a S1000 Thermal Cycler (Biorad, Hercules, CA, USA) and contained 1.5 U of Taq Polymerase (5 U/μL; MP Biomedicals, Santa Ana, CA, USA), 1× PCR Direct Loading Buffer (MP Biomedicals, Santa Ana, CA, USA), 0.4 μM of each pair of primers [53], 200 μM of dNTPs (Jena Biosciences, Jena, Germany), 0.48 μg/μL of BSA (Sigma, Darmstadt, Germany), and 2.5 μL of DNA sample in a final volume of 25 μL. Thermal conditions were 94 °C for 2 min, 35 cycles of 94 °C for 30 s, 61.8 °C for 45 s, and 72 °C for 2 min, and a final step of 72 °C for 7 min. We checked the length of PCR products (i.e., 218 pb) in a 1.2% agarose gel electrophoresis stained with SYBR Safe DNA Stain (Edvotek, Washington, DC, USA). For each round of PCR, we ran negative controls in parallel to detect possible contaminations.

#### 2.5.3. Parasite Loads

To further investigate the degree of infection of bumblebees, we assessed the number of spores present in the gut of bumblebees that were positive for *N. ceranae* in the PCR. We checked spores in the fraction of gut homogenates that was not vortexed. We counted the number of spores under a light microscope (×400; Leica, Wetzlar, Germany) using a Neubauer chamber (Figure 1B). We also screened random samples of *Nosema*-negative PCR bumblebees to confirm the absence of parasite spores.

### 2.6. Experiment 2. Do Bumblebees Evacuate N. ceranae Spores through Faeces?

During infection, *N. ceranae* multiplies in the epithelial cells of the bees and new spores are released with the faeces, becoming a source of infection for new hosts [58]. Thus, the presence of *N. ceranae* spores in the faeces is a proxy for parasite infection. In order to evaluate whether bumblebees evacuate *N. ceranae* spores, we screened the faeces of bumblebees exposed to the parasite for 21 days. We exposed bumblebees to 300 K *N. ceranae* spores (exposed group, *n* = 34), or sucrose solution (control group, *n* = 33) in a 20 μL drop, as in experiment 1. Here, we focused on the highest spore dose because it yielded the highest parasite prevalence in experiment 1 (Figure 1). We then used the bumblebees that drank the full drop (32 and 33 bees, respectively) to create 4 experimental groups: 2 groups in which bumblebees were fed the low-protein diet (N_exposed_ = 16, N_control_ = 19), and 2 groups in which bumblebees were fed the high-protein diet (N_exposed_ = 16, N_control_ = 14). We maintained the bumblebees as described in experiment 1, renewed the diets, and collected the faeces daily. For that, we rinsed each Petri dish with 500 μL of sterile distilled water that we recovered later with a pipette and transferred it to a 1.5 mL Eppendorf tube. We stored the faeces samples at −20 °C until later analyses. When depositions were not found, we forced bumblebees to defecate by slightly pressing their abdomen with forceps. Immediately after each collection, we replaced the Petri dish with a clean one. Before any screening, we centrifuged each sample at 5000× *g* for 5 min, removed the supernatant, and adjusted the final volume to 200 μL with dH_2_0, thus standardising the volume of the samples. Next, we screened the samples in a light microscope (Leica DM750, Germany) and, if found, we counted *N. ceranae* spores in a Neubauer chamber. We then calculated the total number of spores released by bumblebees in 24 h by multiplying the number of spores/μL by the volume of the sample (i.e., 200 μL).

### 2.7. Experiment 3. Does N. ceranae Invade Bumblebee Gut Cells?

Previous descriptions of *N. ceranae* in bumblebees are based on PCR diagnosis or identification of spores in gut homogenates (i.e., [27], this study), but there is still an absence of histological evidence confirming the establishment of the parasite inside the bumblebee gut epithelium [34]. We addressed this question by exposing bumblebees to 300 K spores. We then kept the bumblebees in microcolonies of 25 individuals in an experimental room set at 26 °C, where they fed the low-protein diet. Keeping bees in groups increases the chances that they get infected [59]. We focused on the low-protein diet on the basis of the observation that spores were found in faeces of bumblebees up to 19 days after exposure (experiment 2). By days 2, 9, and 16 after exposure, we sampled 2 exposed bumblebees and 2 controls for later analyses.

Histology samples were prepared as described elsewhere [6,60]. We dissected the digestive canal of each bumblebee with metallic forceps and separated the ventriculus from the rest of the tissue with a scalpel. We fixed each ventriculi in 4% formaldehyde (PanReac Applichem, Germany) for an overnight incubation (~18 h) at 4 °C. We then performed 3 washes on each tissue with phosphate buffer saline (PBS; 1×) for a total time of 3 h and kept the tissues in 70% ethanol. We embedded the tissues in paraffine and cut them in sections of 5 μm in a microtome (Microm HM355; Thermo Scientific, Waltham, MA, USA). We stained sections with haematoxylin and eosin that we later checked in a light microscope (Leica, Wetzlar, Germany). We took pictures with Nikon digital camera DXM1200C in a Nikon Eclipse 80i microscope.

### 2.8. Statistical Analyses

We performed all analyses in R v. 1.0.143 (R Development Core Team). Means are shown with standard errors (mean ± SE).

We tested the effects of diet, spore dosage, and their interaction on parasite prevalence (proportion of individuals infected after parasite exposure) using generalised linear models with a binomial error structure (binomial GLMs). We tested the effect of diet, spore dosage, and their interaction on the proportion of bees showing *N. ceranae* spores in their gut using a binomial GLM. We tested the effect of diet, spore dosage, and their interaction on parasite intensity (number of spores per bee) using a negative binomial GLM (because of the large over-dispersion of the data). We compared the parasite loads of individual bees that survived until the end of the experiment (day 21) and those that died before using a negative binomial GLM with a binary predictor (e.g., dead or alive at day 21) as a fixed effect. We tested the effect of diet on parasite intensity in bumblebee faeces using a negative binomial GLM. We fitted GLMs with binomial distribution errors using the *glm* function in the R package “stats”. We fitted GLMs with a negative binomial error distribution using the *glm.nb* function in the R package “MASS” [61]. We tested all the models for interactions among predictors (e.g., diet, dosage, and PCR results) and removed all interactions that did not improve the fitting using the Akaike information criterion (AIC) for model comparison [62].

We analysed the survival of bumblebees using a Kaplan–Meier test curve with the function *survfit* in the R package “Survival” [63]. We analysed the effects of spore dosage, diet, infectious status, and their interactions using Cox proportional hazards regression models (function *coxph* in the R package “Survival”), followed by a Tukey post hoc test to account for pairwise comparisons. In all models, we included colony origin and incubator identity as random factors.

## 3. Results

### 3.1. Experiment 1. Does Diet Influence N. ceranae Infection?

We exposed bumblebees to two doses of *N. ceranae* and provided them two diets differing in their protein to carbohydrate ratio.

#### 3.1.1. Parasite Prevalence

To assess parasite prevalence, we checked the presence of *N. ceranae* DNA in bumblebees by PCR (see example in Figure 1A). In total, 46.15% of the bumblebees exposed to *N. ceranae* were PCR-positive (Table 2), and 83.8% of the honey bees (119 out of 142), thus confirming that *N. ceranae* spores were infective. Note however that a proportion of non-exposed individuals (5.5% bumblebees and 19.8% honey bees) were also PCR-positive. This suggests that our colonies were not entirely free of parasites prior to the experiments or that horizontal cross-contamination occurred during the experiments. These bumblebees were excluded from further analyses.

Diets significantly influenced the proportion of bumblebees that became PCR-positive after exposure to the parasite (Table 2). A significantly larger number of bumblebees became PCR-positive when fed the low-protein diet in comparison with those fed the high-protein diet (GLM_binomial_: estimate = 1.186 (±0.313), z = 3.79, *p* < 0.001), irrespective of spore dosage (GLM_binomial_: estimate = 0.4870 (±0.313), z = 1.503, *p* = 0.133). Interestingly all the bumblebees that were fed the low-protein diet and that survived until day 21 were PCR-positive to *N. ceranae* (six bumblebees for 150 K, nine bumblebees for 300 K). By contrast, the only bumblebee that was fed the high-protein diet and that survived until day 21 (300 K) was PCR-negative.

#### 3.1.2. Parasite Loads

To assess parasite loads, we screened the gut homogenates of dead PCR-positive bumblebees under a microscope for the presence of *N. ceranae* spores (Figure 1B). We observed *N. ceranae* spores in the gut of bumblebees from both diets (Figure 1C–F; see Appendix A for details). Comparing the presence of spores in bumblebees that did not die on the same day yielded information about the dynamics of spore evacuation and production. For the low-protein diet, we observed spores from day 3 to day 20 after exposure with 150 K spores (Figure 1C) and from day 4 to day 20 after exposure with 300 K spores (Figure 1D). Note, that there was no record of spores between day 8 and day 15, probably because of the low mortality rate of bumblebees on these days. For the high-protein diet, we observed spores from day 2 to 8 after exposure with 150 K spores (Figure 1E), and from day 3 to day 9 after exposure with 300 K spores (Figure 1F).

Overall, 30.9% (26 out of 84) of the PCR-positive bumblebees showed *N. ceranae* spores in their gut (Table 2). This estimation of prevalence based on spore loads was lower than that based on PCR screening (Table 1). This suggests that we may have overlooked the presence of spores in samples with very low infection rate, as well as other intracellular stages of the parasite (e.g., meronts, sporonts) only detected by PCR. The proportion of bumblebees with spores in their gut was similar regardless of the diet (GLM_binomial___diet_: estimate = 0.176 (±0.489), z = 0.36, *p* = 0.718) and the spore dosage (GLM_binomial___spore dosage_: estimate = 0.109 (±0.477), z = 0.23, *p* = 0.819). Spore loads were highly variable across individuals, with an average of 183,655.8 (±36,017.48) spores per bumblebee. The spore dosage the bumblebees were exposed to had no effect on the spore loads found in their gut (GLM_negative-binomial_: estimate = −0.002 (±0.002), z = −0.73, *p* = 0.464). On the contrary, the diet had a significant effect on spore load (GLM_negative-binomial_: estimate = 1.294 (±0.347), z = 3.73, *p* < 0.001), leading to higher amounts of spores in bumblebees fed the high-protein diet (331,250 ± 104,750.4 spores per bee) than in bumblebees fed the low-protein diet (91,406.25 ± 22,851.56 spores per bee).

Considering bumblebees that were fed the low-protein diet and that survived until day 20 post exposure, only four out of six individuals exposed to 150 K spores (75,000 ± 26,615.5 spores per bee; Figure 1C) and four out of nine individuals exposed to 300 K spores (62,500 ± 31,250 spores per bee; Figure 1D) showed spores in their gut. These bumblebees showed similar spore loads than those fed the same diet but that died before day 20 post-exposure (GLM_negative-binomial:_ estimate = −0.363 (±0.389), z = −0.93, *p* = 0.350).

#### 3.1.3. Survival

To test the effect of spore dosage and diet on the survival of bumblebees, we analysed their longevity during the 20 days after parasite exposure. The infection status of each bumblebee was based on the PCR results. Overall, bumblebees fed the high-protein diet had lower survival than bumblebees fed the low-protein diet (Figure 2; Cox: estimate = −0.75 (±0.14), z = −5.17, *p* < 0.001). Bumblebees fed the high-protein diet (Figure 2A) showed similar mortality rate irrespective of spore dosage (Table 3) and infectious status (Appendix A). By contrast, bumblebees fed the low-protein diet (Figure 2B) had a significantly reduced survival when exposed to *N. ceranae* spores than when non-exposed (Table 3). For this diet, 50% of exposed bumblebees died between days 6 and 7 post-exposure. This lethal time 50 (LT_50_) was not reached for the non-exposed (control) bumblebees. Interestingly, in the low-protein diet, PCR positive bumblebees tended to die faster than control bumblebees but survived significantly longer than PCR negative bumblebees (Figure 2B; Table 3). Therefore, both parasite exposure and infection (as measured by PCR) had negative effects on the survival of bumblebees maintained on the low-protein diet.

### 3.2. Experiment 2. Do Bumblebees Evacuate N. ceranae Spores through Faeces?

Intestinal pathogens are released with the faeces, which then constitute a source of infection for new hosts. Here, we investigated the presence of *N. ceranae* spores in bumblebee faeces as evidence of parasite infection. We collected the faeces of each bumblebee at least once, except for one exposed bumblebee that never had faeces spots in the Petri dish and never defecated after pressing his abdomen with forceps. This particular bumblebee was not considered for the analyses (*n* = 31). We screened a subset of faeces from control bumblebees on days 1, 13, 17, 19, and 21 post-exposure to verify the absence of spores (Table 4). We performed more screenings from day 13 post-exposure as we believe that the appearance of any spores in faeces after 10 days post exposure would reflect the releasing of new spores following parasite multiplication, and thus the success of the infection.

We did not find spores in any of the samples analysed from control bumblebees. Likewise, 32.22% of exposed bumblebees (10 out of 31) never showed spores, two of which survived the experiment. By contrast, we found spores in the faeces of 67.74% (21 out of 31) exposed bumblebees at least once (10 fed low-protein diet and 11 fed high-protein diet, Table 4). We detected spores as early as 1 day post-exposure and until day 19 in the low-protein diet but only until day 7 post-exposure in the high-protein diet, even if few bumblebees survived until the end of the experiment.

The number of spores excreted per 24 h in the faeces ranged from 500 to 85,000 (17,462.96 ± 4165.03) and did not differ between bumblebees fed the low-protein diet and those fed the high-protein diet (20,500 ± 7666.73 vs. 14,642 ± 3906.33; GLM_negative-binomial_: estimate = −0.336 (±0.466), z = −0.721, *p* = 0.471). The number of times we found spores in the faeces of the same bumblebee ranged from one to three. Sixteen bumblebees showed spores only once, at days 1 (*n* = 9; 28,333 ± 5830.95 spores), 4 (*n* = 5; 1250 ± 322.74 spores), 7 (*n* = 1; 10,000 spores), and 19 post-exposure (*n* = 1; 50,000 spores). Identification of spores 24 h after feeding the parasite suggests that spores were not retained in the gut and therefore are not the result of an infection [34]. The other five spore-positive bumblebees showed spores at least twice. The two bumblebees fed the high-protein diet presented spores in their faeces on days 1 and 4 or 5 post-exposure, had a 2.5- and 3.3-fold more spores in the second sampling (i.e., 2000 to 5000, and 6000 to 20,000 spores, respectively). Likewise, a bumblebee fed the low-protein diet showed 4.25-fold more spores on day 19 (i.e., 85,000 spores) than on day 15 (20,000 spores). On the contrary, two bumblebees fed the low-protein diet showed a reduction on spore loads by 0.25-fold between the two sampling days (i.e., days 1 and 4, 4000 to 1000 spores; days 4 and 19, 2000 to 500 spores, respectively). Evidence of spores twice in the same individual, after several days of exposure and with an increase in the number of spores relative to the first sample analysed, suggests that the parasite was able to multiply.

### 3.3. Experiment 3. Does N. ceranae Invade Bumblebee Gut Cells?

In honey bees, *N. ceranae* invades the epithelial cells where it multiplies [6]. Given our observation of spores in bumblebee guts and faeces 20 days after parasite exposure, we performed histological analysis to identify parasitic forms inside the epithelial cells of the ventriculi of bumblebees as evidence of parasite infection. We analysed six bumblebees exposed to 300 K spores of *N. ceranae* (two bumblebees at days 2, 9, and 16 post exposure) and three unexposed controls, all maintained on a low-protein diet. We did not find any evidence of parasite infection in any of these bumblebees (see examples in Figure 3). Epithelium of exposed bumblebees did not show signs of damage (i.e., lyses of epithelial cells) and there were no parasitic forms inside the cells.

## 4. Discussion

Several recent studies suggest that *N. ceranae* negatively affects bee behaviour and cognition, with dramatic consequences for colony growth and survival [23,24]. In bumblebees, these results remain difficult to interpret due to the low efficiency of experimental infection protocols. Here, we exploited recent insight into the nutritional ecology of bees [64,65,66] to develop a method to infect bumblebees with *N. ceranae*. We identified key effects of diet composition on bumblebee infection rates, survival, and spore production, indicating a complex interaction between diet, parasite development, and host health.

Overall, 46% of the bumblebees exposed to *N. ceranae* spores were PCR-positive, a percentage that increased to up to 70% when only considering bumblebees that were exposed to 300 K spores and maintained on a low-protein, high-carbohydrate diet. These levels of prevalence are similar to those obtained by Graystock et al. [27] (62% infection rate) with a 20 to 45 times lower infection dose (measured in naturally infected colonies: 6500 spores/bee), and Botías et al. [35] (66% infection rate) with a dose of 130,000 spores per bee. In both studies, the authors administered the parasite with a micropipette directly into the bee mouth parts. These rates of prevalence are higher than all previous studies where bumblebees were exposed to an order of at least 10^5^ spores per bee, either delivered as a drop of solution in a Petri dish [39] or directly fed to bumblebees [23,33,34].

We did not find any clear effect of parasite dosage (150 K or 300 K spores per bee) on prevalence and survival. Comparing our results to that of previous studies does not indicate such effect either, which suggests that above a certain concentration of parasite spores (e.g., 6500 spores/bee in [27]), parasite dosage is not an important factor determining the success of infection. In contrast, we found a strong effect of diet on parasite prevalence and spore loads in bumblebee guts. Pollen intake affects honey bee physiology and tolerance to *N. ceranae* [67,68]. Using artificial diets with controlled amounts of protein and carbohydrates, we demonstrated that food macronutrient balance substantially influences both the infection rate and the survival of bumblebees. More bumblebees were PCR-positive when fed the low-protein, high-carbohydrate diet, suggesting that different blends of nutrients differently affect host susceptibility or parasite physiology (but see [46]). The digestion of the different macronutrients in the bumblebee gut, whose microbiota is mostly composed of sugar-fermenting bacteria [69], may lead to changes in the gut environment that could favour the parasite germination, such as shifts in pH [70]. Furthermore, because *N. ceranae* exploits the metabolism of carbohydrates to obtain energy from its host [8], a higher carbohydrate consumption by infected bumblebees may benefit the establishment of *N. ceranae*. Like previous studies on honey bees that showed that the supplement of pollen in diet increases parasite intensity [46,68,71], we found that bumblebees with access to higher amounts of protein show higher spore loads in their gut. This suggests that proteins benefit the replication of the parasite.

The effect of diet was also evident in bumblebee survival. *N. ceranae* reduced the survival of bumblebees, specifically in the low-protein, high-carbohydrate diet, where the mortality rate increased by day 3 post-exposure. This result is consistent with the time described for *N. ceranae* to produce new spores in the honey bee gut [6]. Similarly to Graystock et al. [27], we found that most of the bumblebees that tested positive for *N. ceranae* in PCR (67.5%) died without showing spores in their gut, suggesting that the parasite was not able to complete its cycle at a detectable level (i.e., produce new spores). The absence of parasite spores in the gut of bumblebees that died at different times could reflect the ability of bumblebees to clear the parasite. The fact that *Nosema*-positive bumblebees (i.e., PCR) fed the low-protein, high-carbohydrate diet survived longer than exposed non-positive bumblebees fed the same diet suggests that the mechanisms allowing parasite clearance incur a higher metabolic cost, here manifested as a reduction of bumblebee lifespan. This cost can be linked to the activation of the immune response against *N. ceranae*, in which several immune-related genes are activated in bumblebees when fed this parasite [35]. Alternatively, interactions between bumblebees and *N*. *ceranae* are recent or rare, and parasites were only able to establish themselves in hosts that for an unknown reason had greater longevity.

We found spores in the faeces of bumblebees fed *N. ceranae*. Some bumblebees only excreted spores with their faeces once, and on the following day after feeding the parasite. Recently, Gisder et al. [34] showed that *N. ceranae* spores travel faster in the digestive tract of bumblebees than in honey bees, which could prevent infection in bumblebees. It is then possible that these spores, ingested by bumblebees at the beginning of the experiment, were fast released and did not produce any infection, as no other spores were observed in samples from the same bumblebees later in the experiment. However, when the excretion of spores happens later in the experiment, i.e., from day 4 or afterwards, it may indicate a possible establishment of *N. ceranae* in the bumblebees and its multiplication, especially in those bumblebees where there was an increase in the number of spore counts (i.e., 20,000 vs. 85,000 spores in days 15 and 19, respectively).

Surprisingly, in the histological analyses, we did not find evidence of parasite infection in the bumblebee ventriculi nor the Malpighi tubules. This suggests that *N. ceranae* does not invade the gut epithelium in bumblebees as it does in *A. mellifera*, where it presents a high tropism [51]. Alternatively, *N. ceranae* might invade other tissues where it could develop. Graystok et al. [27] found molecular evidence for the presence of *N. ceranae* in the fat body of bumblebees, suggesting the migration of *N. ceranae* from the gut to this organ [27]. The fat body is also invaded by the microsporidia *Nosema bombi* in bumblebee infections [72,73]. However, further analyses are needed to test this hypothesis.

The fact that bumblebees showed low parasite prevalence but died faster when fed a high-protein, low-carbohydrate diet suggests that immunity and lifespan are maximised at different nutritional balances. Several studies show how an animal’s diet can differently influence the expression of key life-history traits, forcing the animal to the trade-off between optimising multiple traits simultaneously (e.g., [74,75,76]). In insects, lifespan is typically enhanced on high-carbohydrate diets whereas reproduction is maximised on high-protein diets (fruit flies [77,78,79,80,81], crickets [82]). Immunity and reproduction also display differences in nutritional requirements (as shown in fruit flies [40,83], decorated crickets [76], cotton leafworm [43]). Honey bees survive longer on high-carbohydrate diets [84], and pollen (the main source of protein) favours the survival of individuals infected with *N. ceranae* [46,68,71,85]. Whether sterile bumblebee workers trade-off between over-ingesting protein and under-ingesting carbohydrates in order to reduce parasite establishment at the expense of a shorter lifespan is an open question. In these social insects, such individual strategy may reduce the risks of contamination of other workers within the colony [50].

Developing standard protocols to characterise the sublethal effects of parasites and pathogens on bees has become a major challenge for understanding bee population declines [4]. Here, we demonstrated that diet is key in determining infection rates of bumblebees exposed to *N. ceranae*. The highest infection rates and the longest survivals were obtained with a high-carbohydrate, low-protein diet, thereby providing ideal conditions for investigating the potential effects of *N. ceranae* on bee behaviour and cognition. Other parameters, not tested here, may also be of importance and investigated in future studies. For instance, both our study and Graystock et al. [27] starved bumblebees for at least 5 h before to parasite exposure, which may have increased the probability of parasite establishment. Variations in the age of the bumblebee tested may also explain some of these differences. In the only study that controlled for age, Fürst et al. [39] infected 2-day-old bumblebees (post-eclosion from the pupa) and obtained a lower infection rate than the one obtained in this study and found no effect on bumblebee mortality, suggesting that young bumblebees are less susceptible to the parasite. Finally, we cannot exclude differences in the virulence of the *N. ceranae* strains used for experimental infection across studies (but see [86]). We hope that our infection protocol will stimulate further studies on the interactions between *N. ceranae* and bumblebees in order to better assess the risks this emergent parasite represents for wild pollinators.

## Figures and Tables

**Figure 1 microorganisms-09-00158-f001:**
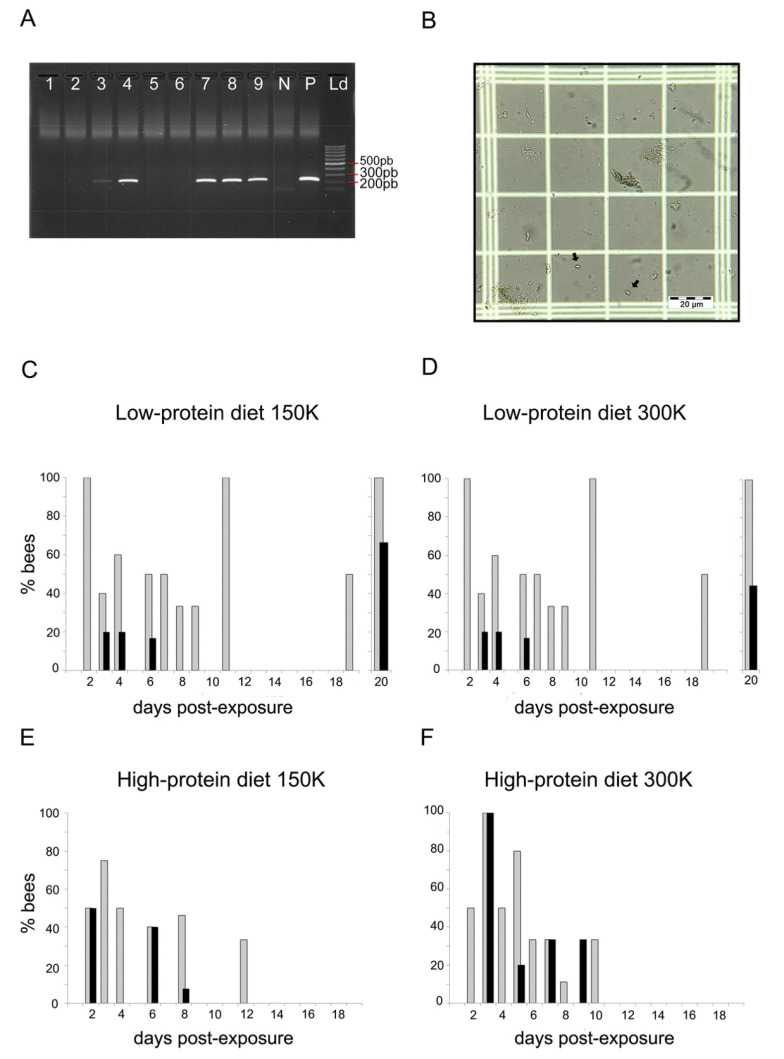
Parasite loads. (**A**) Example of agarose electrophoresis gel with negative (lanes 1, 2, 5, and 6) and positive (lanes 3, 4, 7–9) bumblebees to *N. ceranae* determined by PCR. N: negative control of PCR; P: positive control of PCR; Ld: molecular size marker (100 pb). (**B**) Example of *N. ceranae* spores observed in an infected bumblebee (optical microscope, ×400). (**C**–**F**) Proportion of dead bumblebees at different days postexposure that were PCR-positive to *N. ceranae* (grey bars) and showed spores in their gut (black bars) for both diets (high- and low-protein diets) and spore dosages (150 K and 300 K). Each group contained between 44 and 47 individuals.

**Figure 2 microorganisms-09-00158-f002:**
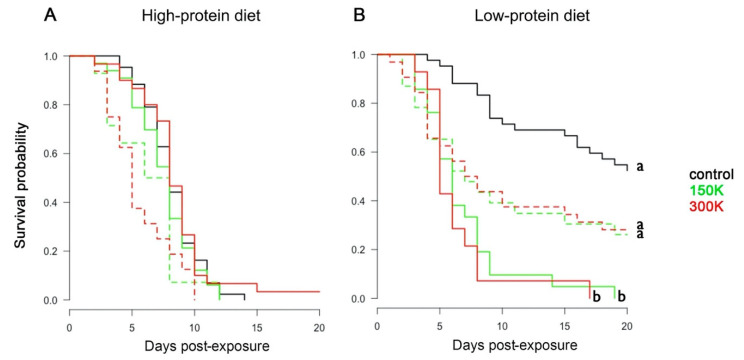
Survival analyses. Survival probability of bumblebees fed the high-protein diet (**A**) and the low-protein diet (**B**) for the different spore dosages (control, 150 K, and 300 K) across time. PCR-positive (dashed lines) and PCR-negative (solid lines) bumblebees are differentiated. Different letters associated with curves indicate statistical differences in survival (*p* < 0.05; post hoc Tukey test after Cox regression model in a Kaplan–Meier test curve, see Appendix A). Each group contained between 44 and 47 individuals.

**Figure 3 microorganisms-09-00158-f003:**
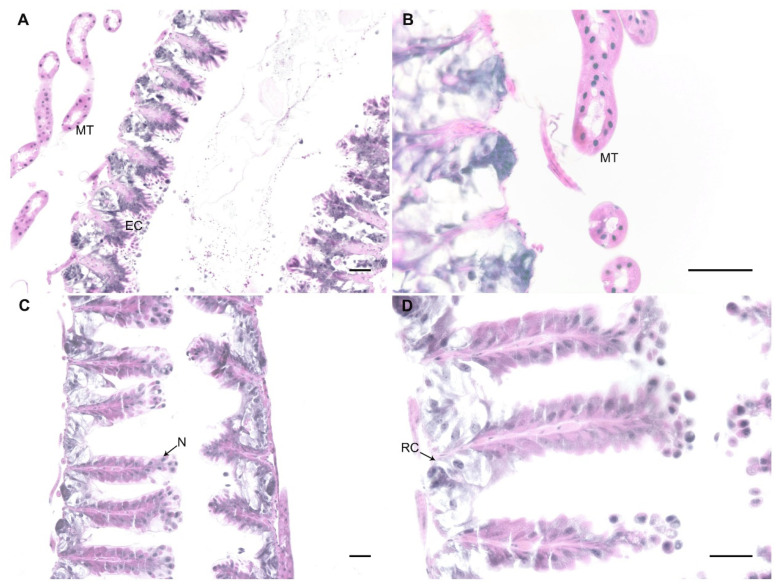
Examples of midgut sections under the light microscope of a portion of the ventriculus of control (**A**) and exposed bumblebees at days 9 (**B**) and 16 (**C**,**D**) post-exposure. No evidence of parasitic form was observed in the epithelium (EC), nor in the Malpighian tubules (MT). N: nuclei, RC: regenerative cells. Haematoxylin and eosin staining. Bar scale = 25 µm.

**Table 1 microorganisms-09-00158-t001:** Summary of protocols developed to infect bumblebees (*Bombus terrestris*) with *Nosema ceranae* spores.

Spores per Bumblebee	Starvation Duration	Age of Bumblebee	Parasite Exposure	Experimental Condition	Rate of Infection (Based on PCR)	Study
120,000	4 h	Unknown	Micropipette 30% sugar-water	Full colony; first 2 weeks: 50% sucrose solution + pollen 25 °C, 50–60% RH Later on: field	66%	[35]
6500	8 h	Unknown	Hand-fed; 40% sucrose	Groups of 10 40% sucrose solution N/A	62%	[27]
100,000	30–60 min	2 days old	Individually in a Petri dish 50% sucrose solution	Individually 50% sucrose solution + artificial pollen N/A	34% No effect on survival	[39]
50,000	4 h	Newly hatched	Fed individually Sucrose/pollen solution	Groups of 10–25; 50% (*w*/*v*) sucrose + 15% (*w*/*v*) pollen 33 °C and 55% RH	6.13%	[34]
180,000	2 h	Unknown	Inoculated individually Sucrose (30%)	Groups of 10; 50% sucrose solution + artificial pollen 25 °C and 50% RH	3%	[23]
6500–5,000,000	4 h	Newly hatched, 4 weeks old or mixed	Fed individually Sucrose-/pollen solution	Groups of 20; 50% (*w*/*v*) sucrose + 15% (*w*/*v*) pollen 33 °C and 55% RH	0%	[34]
130,000	2 h	Unknown	Micropipette Sucrose (30%)	Microcolonies of 10; 60% sucrose solution + artificial pollen 26 °C, 55% RH	0%	[33]

**Table 2 microorganisms-09-00158-t002:** Number of bumblebees exposed to *N. ceranae* that were PCR-positive and showed spores in their gut. The percentage of PCR-positive bumblebees relative to all exposed bumblebees and the percentage of bumblebees showing spores in their gut relative to all PCR-positive individuals are given in brackets.

Diet	Spore Dosage	Exposed	PCR-Positive	Showing Spores
Low-protein	150 K	44	23 (52.27%)	7 (30.4%)
	300 K	45	31 (68.88%)	9 (29.03%)
High-protein	150 K	47	14 (29.78%)	4 (28.5%)
	300 K	46	16 (34.78%)	6 (37.5%)
Total		182	84 (46.15%)	26 (30.9%)

**Table 3 microorganisms-09-00158-t003:** Cox regression model testing the effect of spore dosage on the survival of bumblebees for each diet.

Diet	Spore Dosage	Estimate	SE	z	*p*
High-protein	150 K	0.2766	0.2115	1.308	0.191
	300 K	0.1440	0.2151	0.669	0.503
Low-protein	150 K	1.2595	0.2792	4.511	<0.001
	300 K	1.1021	0.2808	3.925	<0.001

**Table 4 microorganisms-09-00158-t004:** Number of faeces screened from exposed and control bumblebees in each diet at different days post-exposure. Number of faeces samples showing spores is shown in parentheses.

	Days Post-Exposure											Diet
	1	4	5	7	9	11	13	15	17	19	21	
Exposed	6(4)	14(5)	14(0)	12(0)	11(0)	9(0)	11(0)	6(1)	6(0)	4(3)	2(0)	Low P
Control	2	0	0	0	0	0	13(1)	0	8	7	7	
Exposed	8(8)	14(3)	15(2)	12(1)	12(0)	9(0)	8(0)	8(0)	7(0)	5(0)	0	High P
Control	1	0	0	0	0	0	0	0	3	2	2	

## Data Availability

Data is contained within the article or as Appendix A.

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
