# Peer review of "Artificial Diets Modulate Infection Rates by Nosema ceranae in Bumblebees"

_microorganisms, 2021, doi:10.3390/microorganisms9010158_

Round 1
Reviewer 1 Report
The paper entitled: Artificial diets modulate infection rates by Nosema ceranae in bumblebees by Tamara Gómez-Moracho, Tristan Durand, Cristian Pasquaretta, Philipp Heeb and Mathieu Lihoreau, describes an important protocol for infecting bumblebees (Bombus terrestris) with controlled amounts of N. ceranae, by exposing bumblebees to parasite spores while feeding them with different protein/carbohydrate ratio in artificial diets.
Developing standard protocols is a foundation for the research, ensuring that studies performed by different laboratories would be comparable.
I recommend the manuscript for publication after minor revision: please correct the references in chapter “Results” where the statement “Error! Reference source not found.” is repeating.
Author Response
Thank you for your comments. The error messages concerned references to Tables and Figures. This is now fixed.
Reviewer 2 Report
Comments of the reviewer
Page 1, line 15: remove an before energetic
Page 1, line 26: remove in before the faces
Page 1, line 27: add the before highest
Page 1, line 28: add the before longest
Page 1, line 29: replace had no effect with did not affect
Page 1, line 40: add comma before and
Page 2, line 47: remove an before energetic
Page 2, line 48: remove an before increased
Page 2, line 49: remove an before inhibition
Page 2, line 50: remove an before deregulation
Page 2, line 50: add comma before and
Page 2, line 57: add comma before and
Page 2, line 58: add the before contamination
Page 2, line 68: replace infections with infection
Page 2, line 77: replace prior to with before
Page 2, line 78: remove to after exposed
Page 2, line 77: replace amount with number
Page 2, line 53: add comma before and
Page 4, line 102: remove space before point
Page 4, line 101: replace indicate with indicates
Page 4, line 100: replace in order to with to
Page 4, line 114: add comma before and
Page 4, line 114: add the before best
Page 4, line 116: replace 21-day period with 21 days
Page 4, line 116: remove an before evidence
Page 4, line 124: replace prior to with before
Page 5, line 159: add comma after bees
Page 5, line 168: replace consisted in with consisted of
Page 5, line 170: replace consisted in with consisted of
Page 5, line 173: add comma before and
Page 5, line 175: remove to before 300,000
Page 5, line 181: add a before fresh
Page 6, line 194: remove comma before and
Page 6, line 209: replace positive to with positive for
Page 6, line 219: replace during 21 days with for 21 days
Page 6, line 222: add the before highest
Page 6, line 231: replace by a clean with with a clean
Page 7, line 240: add an before absence
Page 7, line 247: add comma before and
Page 7, line 247: add comma after exposure
Page 7, line 254: replace cut in sections with cut them in sections
Page 7, line 261: add comma before and
Page 7, line 270: add a before fixed
Page 7, line 279: add comma before and
Page 8, lines 312-325: Error! Reference source not found.
Page 8, line 298: Error! Reference source not found.
Page 8, lines 289-290: Error! Reference source not found.
Page 8, line 304: add comma before and
Page 8, line 307: remove comma before and
Page 8, line 306: add the before number
Page 8, line 312: add a before microscope
Page 9, line 341: Error! Reference source not found.
Page 9, line 342: Error! Reference source not found.
Page 10, line 347: add comma before and
Page 10, line 350: add the before proportion
Page 10, line 351: replace were with was
Page 11, line 368: Error! Reference source not found.
Page 11, line 361: Error! Reference source not found.
Page 11, line 362: Error! Reference source not found.
Page 11, line 358: Error! Reference source not found.
Page 11, line 376: associated to replace with associated with
Page 12, line 389: Error! Reference source not found.
Page 12, line 397: Error! Reference source not found.
Page 12, line 382: replace that then with which then
Page 12, line 384: remove an before evidence
Page 12, line 412: replace at day with on day
Page 12, line 413: replace at day with on day
Page 12, line 425: remove an before evidence
Page 12, line 424: remove a before histological
Page 13, line 428: Error! Reference source not found.
Page 13, line 432: add the before light microscope
Page 13, line 433: replace no evidences of parasitic forms were with no evidence of parasitic forms was
Page 13, line 442: add comma before and
Page 13, line 450: add comma after studies
Page 14, line 458: remove an before infection
Page 14, line 456: replace neither with either
Page 14, line 466: replace composed by with composed of
Page 14, line 474: replace evident on with evident in
Page 14, line 476: add the before mortality
Page 14, line 478: replace positive to with positive for
Page 14, line 492: replace in the following with on the following
Page 14, line 496: replace did not produced with did not produce
Page 14, line 502: add comma after analyses
Page 14, line 502: replace evidences with evidence
Page 14, line 503: remove in after nor
Page 15, line 516: insert closing parenthesis
Page 15, line 518: replace leafworm with cotton leafworm
Page 15, line 513: add the before trade-off
Page 15, line 527: add the before highest
Page 15, line 529: add the before potential
Page 15, line 531: replace prior with before
Page 15, line 543: add the before number
Page 16, line 571: insert number 4 in front of the reference
Author Response
Thanks for your comments, please find below the list of changes made
Page 1, line 15: remove an before energetic (Done)
Page 1, line 26: remove in before the faces (Done)
Page 1, line 27: add the before highest (done)
Page 1, line 28: add the before longest (done)
Page 1, line 29: replace had no effect with did not affect (done)
Page 1, line 40: add comma before and -> (we didn’t make the change after careful consideration)
Page 2, line 47: remove an before energetic(done)
Page 2, line 48: remove an before increased (done)
Page 2, line 49: remove an before inhibition (done)
Page 2, line 50: remove an before deregulation (done)
Page 2, line 50: add comma before and (done)
Page 2, line 57: add comma before and -> (we didn’t make the change after careful consideration)
Page 2, line 58: add the before contamination (Done)
Page 2, line 68: replace infections with infection (done)
Page 2, line 77: replace prior to with before (done)
Page 2, line 78: remove to after exposed (done)
Page 2, line 77: replace amount with number (done)
Page 2, line 53: add comma before and (done) -> (Done)
Page 4, line 102: remove space before point (done)
Page 4, line 101: replace indicate with indicates (done)
Page 4, line 100: replace in order to with to (done)
Page 4, line 114: add comma before and (done)
Page 4, line 114: add the before best (done)
Page 4, line 116: replace 21-day period with 21 days (done)
Page 4, line 116: remove an before evidence (done)
Page 4, line 124: replace prior to with before (done)
Page 5, line 159: add comma after bees -> (we didn’t make the change after careful consideration)
Page 5, line 168: replace consisted in with consisted of (done)
Page 5, line 170: replace consisted in with consisted of (done)
Page 5, line 173: add comma before and -> (Done)
Page 5, line 175: remove to before 300,000 (done)
Page 5, line 181: add a before fresh (done)
Page 6, line 194: remove comma before and -> (we didn’t make the change after careful consideration)
Page 6, line 209: replace positive to with positive for (done)
Page 6, line 219: replace during 21 days with for 21 days (done)
Page 6, line 222: add the before highest (done)
Page 6, line 231: replace by a clean with with a clean (done)
Page 7, line 240: add an before absence (done)
Page 7, line 247: add comma before and (done)
Page 7, line 247: add comma after exposure (done)
Page 7, line 254: replace cut in sections with cut them in sections (done)
Page 7, line 261: add comma before and (done)
Page 7, line 270: add a before fixed (done)
Page 7, line 279: add comma before and (done)
Page 8, lines 312-325: Error! Reference source not found (corrected).
Page 8, line 298: Error! Reference source not found.(corrected)
Page 8, lines 289-290: Error! Reference source not found (corrected).
Page 8, line 304: add comma before and (done)
Page 8, line 307: remove comma before and (done)
Page 8, line 306: add the before number -> (we didn’t make the change after careful consideration)
Page 8, line 312: add a before microscope (done)
Page 9, line 341: Error! Reference source not found (corrected).
Page 9, line 342: Error! Reference source not found (corrected).
Page 10, line 347: add comma before and -> (we didn’t make the change after careful consideration)
Page 10, line 350: add the before proportion -> (we didn’t make the change after careful consideration)
Page 10, line 351: replace were with was -> (we didn’t make the change after careful consideration)
Page 11, line 368: Error! Reference source not found. (corrected)
Page 11, line 361: Error! Reference source not found. (corrected)
Page 11, line 362: Error! Reference source not found. (corrected)
Page 11, line 358: Error! Reference source not found.(corrected)
Page 11, line 376: associated to replace with associated with (done)
Page 12, line 389: Error! Reference source not found.(corrected)
Page 12, line 397: Error! Reference source not found.(corrected)
Page 12, line 382: replace that then with which then (done)
Page 12, line 384: remove an before evidence (done)
Page 12, line 412: replace at day with on day (done)
Page 12, line 413: replace at day with on day (done)
Page 12, line 425: remove an before evidence (done)
Page 12, line 424: remove a before histological (done)
Page 13, line 428: Error! Reference source not found.(corrected)
Page 13, line 432: add the before light microscope (done)
Page 13, line 433: replace no evidences of parasitic forms were with no evidence of parasitic forms was (done)
Page 13, line 442: add comma before and (done)
Page 13, line 450: add comma after studies (done)
Page 14, line 458: remove an before infection (done)
Page 14, line 456: replace neither with either (done)
Page 14, line 466: replace composed by with composed of (done)
Page 14, line 474: replace evident on with evident in (done)
Page 14, line 476: add the before mortality (done)
Page 14, line 478: replace positive to with positive for (done)
Page 14, line 492: replace in the following with on the following (done)
Page 14, line 496: replace did not produced with did not produce (done)
Page 14, line 502: add comma after analyses (done)
Page 14, line 502: replace evidences with evidence (done)
Page 14, line 503: remove in after nor (done)
Page 15, line 516: insert closing parenthesis (done)
Page 15, line 518: replace leafworm with cotton leafworm (done)
Page 15, line 513: add the before trade-off (done)
Page 15, line 527: add the before highest (done)
Page 15, line 529: add the before potential (done)
Page 15, line 531: replace prior with before (done)
Page 15, line 543: add the before number -> (we didn’t make the change after careful consideration)
Page 16, line 571: insert number 4 in front of the reference (done)